# Volitional Generation of Reproducible, Efficient Temporal Patterns

**DOI:** 10.3390/brainsci12101269

**Published:** 2022-09-20

**Authors:** Yuxiao Ning, Guihua Wan, Tengjun Liu, Shaomin Zhang

**Affiliations:** 1Qiushi Academy for Advanced Studies, Zhejiang University, Hangzhou 310027, China; 2Department of Biomedical Engineering, Zhejiang University, Hangzhou 310027, China; 3Key Laboratory of Biomedical Engineering of Ministry of Education, Zhejiang University, Hangzhou 310027, China; 4Zhejiang Provincial Key Laboratory of Cardio-Cerebral Vascular Detection Technology and Medicinal Effectiveness Appraisal, Zhejiang University, Hangzhou 310027, China

**Keywords:** precise temporal patterns, energy-efficient code, brain–machine interfaces, primary motor cortex

## Abstract

One of the extraordinary characteristics of the biological brain is the low energy expense it requires to implement a variety of biological functions and intelligence as compared to the modern artificial intelligence (AI). Spike-based energy-efficient temporal codes have long been suggested as a contributor for the brain to run on low energy expense. Despite this code having been largely reported in the sensory cortex, whether this code can be implemented in other brain areas to serve broader functions and how it evolves throughout learning have remained unaddressed. In this study, we designed a novel brain–machine interface (BMI) paradigm. Two macaques could volitionally generate reproducible energy-efficient temporal patterns in the primary motor cortex (M1) by learning the BMI paradigm. Moreover, most neurons that were not directly assigned to control the BMI did not boost their excitability, and they demonstrated an overall energy-efficient manner in performing the task. Over the course of learning, we found that the firing rates and temporal precision of selected neurons co-evolved to generate the energy-efficient temporal patterns, suggesting that a cohesive rather than dissociable processing underlies the refinement of energy-efficient temporal patterns.

## 1. Introduction

Although the human brain actively drains 20% of the body’s energy, it ultimately consumes only 20 watts of power [1,2]. Moreover, one of the newest studies has revealed that power expended for neural computation takes less than one percent of the total budget [3]. Compared to the energy consumed by modern artificial intelligence (AI) implemented in silicon-based hardware, the brain consumes extraordinarily little to implement biological intelligence. For decades, scientists and engineers have attempted to draw insights from neural coding schemes in order to build efficient intelligent systems. Temporal coding, by which information is communicated and processed through the temporal coordination of spikes, utilizes the discrete nature of spikes. The temporal code can theoretically carry more information than the rate code [4]. Therefore, temporal coding has attracted attention and has been employed in AI practices [5,6,7]. However, the implementation of temporal codes does not necessarily lead to sparsity or a low energy budget globally in biological brains (Figure 1A). According to [8], code efficiency is defined as the ratio of the representation capacity to the expended energy, which is characterized by the number of action potentials included. One recent study has indicated that the generation of precise temporal patterns is accompanied by elevated excitation [9]. In this case, although precise temporal patterns were robustly generated, they were not considered efficient, owing to the growing metabolic cost brought about by surplus spikes. Therefore, a deeper understanding of how efficient temporal patterns are generated and of the neural difference underlying the generation of efficient and non-efficient patterns is very much needed. This deeper understanding is the key to drawing functional implications in the realm of neuroscience and designing efficient artificial intelligence.

Over the last decades, researchers have been documenting efficient temporal patterns, mainly in the context of stimulus encoding [10,11]. However, in addition to representing stimulus, various computations are performed in the biological brain to yield intelligence, e.g., the working memory, generation of complex self-sustained patterns, and the issuing of outputs. Nevertheless, very little is known about whether the efficient temporal patterns could be reproducibly implemented in order to serve a broader range of functions.

Another controversial topic in relation to efficient code is its learnability. Since the temporal displacement of each spike “counts”, and any surplus spikes would be regarded as metabolically costly by the definition of efficient code, the deployment of this code in the spike-based neural network in order to support efficient computation holds much promise. However, relatively few studies have demonstrated that networks could be trained to generate efficient patterns [12,13,14]. One proposed strategy was to explicitly constrain the firing rates in the cost function and to train the network with global error back-propagation, which was biologically implausible (Figure 1B). Nevertheless, it has not been demonstrated whether the efficient patterns could be learned in a biologically plausible manner, which could be challenging to investigate in vivo since one cannot define “objectives” to constrain the emission of spikes during traditional behavioral tasks, which is the case in the artificial neural networks.

To address these challenges, we took advantage of a brain–machine interface (BMI) [15]. Aside from its resounding success in restoring motor [16,17,18,19,20], speech [21], and psychological functions [22], BMI has recently energized neuroscience by offering causality to behavioral outcomes for certain neural patterns. Researchers can define the mappings between neural activity and behavioral outcomes, which were termed “decoders” [23,24,25,26]. Thus, BMI endowed us with the ability to define “objectives”, namely to constrain the emission of spikes in order to investigate whether the efficient temporal patterns could be reproducibly generated and learned (Figure 1C). In the study of [9], the authors showed that when the decoders were a function of relative spike timing rather than commonly used descriptors such as firing rates, to control outcomes, the patterns elicited with temporal precision could be causally related to behaviors. Furthermore, known mappings in such “temporal neuroprosthetics” allowed the authors to track how the temporal patterns were learned and refined in the brain circuits. However, that study only covered the case of non-efficient temporal patterns, leaving efficient temporal patterns largely unexplored. Therefore, the focus of this work is to study the generation and the learning of efficient temporal patterns in the motor cortex by introducing a new BMI that constrains the precise spike timing as well as the metabolic cost brought by the spike counts.

## 2. Materials and Methods

All surgical and experimental procedures conformed to the Guide for the Care and Use of Laboratory Animals (China Ministry of Health) and were approved by the Animal Care Committee of Zhejiang University in China. The surgical procedure was described in detail in [27]. Briefly, the 96-channel microelectrode arrays (Blackrock Neurotech) were chronically implanted in the primary motor cortex of two male rhesus monkeys (Macaca mulatta) (Monkey B11, Monkey C05). The monkeys took approximately a week to recover from surgery, after which the neural signals were recorded through the Cerebus multichannel data acquisition system (Blackrock Neurotech) at a sample rate of 30 kHz. Spike activities were detected by thresholding (root mean square multiplier, B11: ×5.5; C05: ×7). In the first learning session and after online manual sorting, we assigned two isolated units with the highest signal-to-ratio (SNR) and waveform stability based on recordings from prior weeks to be the trigger unit and the target unit, respectively. The online sorting templates for these two units were unchanged throughout the learning and validated by the offline sorter (Plexon, Dallas, TX, USA, Inc.) to ensure the same neurons were recorded and used for training across sessions. It is shown in Appendix A that the variation of waveforms across sessions is smaller than the average amplitude. Since these two selected units were directly responsible for the outcome, they were labeled as “direct neurons”. The remainder of the units were “indirect neurons”, which were detected and sorted offline for the subsequent analysis in this article. The number of recorded “indirect neurons” was relatively stable across days (B11: 54.3 ± 8.5; C05: 54.3 ± 9.1). Additionally, the spatial relationship of the direct neurons in the array is shown in Appendix A.

### 2.1. Behavioral Task

In this task, the spikes of the trigger unit and the target unit in a window of 300 ms were streamed in real time to custom-written scripts in MATLAB (Mathworks Inc., Natick, MA, USA) in order to compute a conditioning variable: the normalized coincidence score (NCS). The window was slid every 150 ms with a 150 ms overlap. NCS was then fed back to the subject aurally and visually, every 150 ms as well. The subjects would receive the water reward if they managed modulate the neural patterns to drive the NCS to the threshold in 15 s. Otherwise, the trial would be terminated, and the subjects would have to wait for another 4 s to initiate the next trial (Figure 2B). The NCS thresholds to obtain the reward were set to be the 99th percentile of the NCS distribution, which were estimated from the baseline data in the first session (sampled every 150 ms for 5 min). The NCS thresholds (B11: 0.36; C05: 0.32) were fixed across sessions. In the baseline period, the subjects sat with their hands restricted. Water was given randomly but sparsely, only to calm the subject and to allow them to maintain a stationary state devoid of large movements. No auditory or visual stimuli were given in this period.

The conditioning variable NCS is defined as follows:(1)$$NCS=\frac{\sum_{i=1}^{M}\sum_{j=1}^{N}S\left(\Delta_{ji}\right)}{\sqrt{M\times N}}$$
where *M* and *N* are the total number of spikes from the trigger unit and the target unit in a 300 ms time window, $S(\Delta_{ji})$ is the score function dependent on the lag $\Delta_{ji}$ between the emission time of the $i^{th}$ spike of the trigger unit and the $j^{th}$ spike of the target unit. More specifically, the score function takes on an exponential form, similar to how synaptic efficacy changes under spike timing-dependent plasticity (STDP) [28], where the trigger leading the target results in a positive score and vice versa:(2)$$S\left(\Delta \right)=\left\{\begin{array}{c} e^{-\frac{\Delta }{\tau }}\;\Delta >0\\ -e^{-\frac{\Delta }{\tau }}\;\Delta <0 \end{array}\right.$$In our experiment, τ was set at 17 ms to fit the critical window observed in STDP experiments [29]. Therefore, the numerator of the NCS served to reward not only the temporal precision, but also the correct temporal order. For the sake of convenience, we termed this numerator the coincidence score (CS). Normalization by the geometric mean of the spike counts from the two units played the role of penalizing excessive spikes for constraining energy expenses. Theoretically, the value of NCS should be within the range from −1 to 1.

We mapped the NCS into the frequency of an audio cursor, ranging from 1 kHz to 24 kHz in quarter-octave increments. Moreover, we adopted a similar visual feedback environment in the center-out tasks, since all subjects had been trained on center-out tasks. NCS in [−0.5, 0.5] was mapped to the vertical position of a blue circle on the screen (Figure 2A), with a yellow circle indicating the reward threshold. The visual feedback was implemented using Psychtoolbox and interfaced with the custom-written main program.

The subjects were water-constrained and had to learn tasks on consecutive days, with two sessions being held on each day, one in the morning and another in the afternoon.

We also conducted a control experiment to investigate how the normalization of firing rates impacted the generation of efficient temporal neural patterns. In this control experiment, C05 had to modulate the coincidence score rather than the normalized coincidence score using a different pair of direct units (termed “CS-modulation” task). Other task configurations were identical to those in the “NCS-modulation” task.

### 2.2. Data Analysis

*Behavioral metrics.* We applied two metrics, the success rate and the trial duration, in order to evaluate the behavioral performance in the task. The success rate was defined as the ratio between the number of successful trials and the total number of trials in one session. Relatively the same total number of trials was performed across sessions for each subject (B11: 105.3 ± 12.8; C05: 215 ± 20.0) and was used for the subsequent analysis. The number of trials was set to the amount that subjects could perform with full engagement.

Another metric that assesses behavioral performance, trial duration, was defined as the time from the start of a trial to the reward delivery.

*CCH.* We used a jitter-corrected cross-correlation histogram (CCH) to portray the temporal relationship between the trigger and target units on a fine time scale [30]. We randomly jittered the emission time for each spike from the target unit in order to render the surrogated CCH (sCCH). This resampling procedure was run 1000 times [31]. The variation in jitter was dependent on the temporal resolution of the CCH. For example, if the bin of CCH was 15 ms, as shown in Figure 5A, then the standard deviation of random jittering (“jitter window”) should also be set at 15 ms. We subtracted this sCCH from the raw CCH to obtain the corrected CCH. In this way, we guaranteed that structured firing patterns or variations in timescales comparable to the temporal resolution of CCH would be removed. Essentially, each bar of a particular bin in the CCH represents the possibility of a spike of trigger neurons to generate a coincidence with the target neuron. Therefore, we could use the bar standing beside zero-lag to measure the temporal precision of two neurons. Since our study focused on temporal precision with temporal order, we only used coincidence in [0, 15] ms or [0, 5] ms for analysis. In addition, the tail probability of sCCH was used to construct the acceptance bands. The error bar in the jitter-corrected CCH outlined the standard deviation of the sCCH.

*Comparison of spike counts for precise temporal patterns.* To test the hypothesis that the generated precise temporal patterns are energy efficient, we compared the spike counts of the precise temporal patterns collected from the task block with those from the baseline block while controlling for the coincidence score. More specifically, we sampled the 300 ms long neural patterns from the baseline block (by moving window in 150 ms steps), whose coincidence score was in the range of the coincidence scores from the rewarding neural patterns.

Additionally, we also compared the spike counts of the precise temporal patterns collected from this task with those from the rewarding neural patterns of the CS-modulation task. Similarly, we sampled the neural patterns from these two sets so that the range of coincidence scores would match.

*Modulation index.* The modulation index was used to characterize the modulation depth of single neurons in the task block grounded on its firing in the baseline block $FR_{baseline}$:(3)$$modulation\;index=\frac{FR_{baseline}-FR_{task}}{FR_{baseline}+FR_{task}}$$More specifically, $FR_{task}$ was estimated using the successful trials in the task block.

*Stable sessions.* To find out the stable sessions (or learning plateau) in the total *n* recording sessions, we iteratively computed the ratio $r_{k}$ between the variance of the success rate from the last *k* sessions $\sigma^{2}\left(\left\{S_{i}|i=n-k+1,...,n\right\}\right)$ and that from the first $\sigma^{2}\left(\left\{S_{i}|i=1,...,n-k\right\}\right)$ as follows:(4)$$r_{k}=\frac{\sigma^{2}\left(\left\{S_{i}|i=n-k+1,...,n\right\}\right)}{\sigma^{2}\left(\left\{S_{i}|i=1,...,n-k\right\}\right)}$$
where $S_{i}$ denotes the success rate at session *i*. *k* was iterated from 2 to n − 1, and the iteration was terminated when $r_{k}$ was greater than $r_{k-1}$. The last k−1 sessions would be returned as the stable sessions. In other words, adding the $k^{th}$ session to the last k−1 sessions would enlarge the behavioral variation and thus need not be considered as a stable session. Based on this criterion, session 6 to session 10 of B11 and session 9 to session 10 of C05 were considered stable.

*Overshooting NCS.* The NCS that surpassed the threshold for reward was defined as the overshooting NCS. To counter the disparity of threshold values in the two subjects, we subtracted the difference in thresholds (0.36 − 0.32 = 0.04) from all the overshooting NCS of B11 in order to render the corrected overshooting NCS.

### 2.3. Statistical Analysis

*Success rate.* Linear fitting was first performed to test for the upward trend in each subject. A one-tailed Mann–Whitney test was further used to compare the success rate in the early phase versus that in the late phase. Each group combined the four sessions of the two subjects (n = 8) [32].

*Trial duration.* To compare the trial duration in the early phase with that in the late phase, we pooled together samples of trial duration from different sessions and subjects. Given this nested structure, a two-way ANOVA (*F*(DFn, DFd)) was used to reveal the variation of the “phase” factor and the interaction between the different factors.

*Test for significant coincidence.* The standard deviation of the surrogated CCH after resampling was used to compute the upper limit of the confidence interval for each bin under different alphas (e.g., 0.05 for 95% confidence). We thus tested the significant coincidence of each bin by comparing the jitter-corrected CCH with those upper limits.

*Efficient firing with firing rate normalization.* Given that the spike count in the 300 ms long neural pattern was a discrete variable, the Mack–Skillings test (non-parametric two-way ANOVA) was used to test against the null hypothesis that spike counts of precise temporal patterns from a task block and from a baseline block (or a CS-modulation task) had no difference. With the coincidence score as the second factor, it was binned into five blocks from 0.9 to 2.9 (three blocks from 1.8 to 3.2 when compared to the CS-modulation task).

*Network modulation index.* The modulation indices of all significantly modulated indirect neurons across sessions (four sessions for each condition) were pooled together, and a one-sample t-test was used to examine whether the mean modulation index was statistically different from zero.

*Converged distribution of corrected overshooting NCS.* A two-sample Kolmogorov–Smirnov test was used to test against the null hypothesis that the corrected overshooting NCS of two subjects were from the same distribution.

## 3. Results

We devised a novel BMI paradigm to explore the plausibility of learning and generating reproducible, efficient temporal patterns in M1. The guiding principle in designing the paradigm was to encourage higher temporal precision while penalizing excessive firing. To this end, two key ingredients in a BMI system should be purposely designed: the “decoder” and the feedback approach. For the “decoder”, we developed the normalized coincidence score (NCS) as the decoded variable. The subjects needed to up-modulate the NCS to surpass a threshold in order to obtain rewards (Figure 2). Furthermore, unlike the widely used BMI “decoders” based on linear time-invariant mappings on firing rates in a single timescale, ours was based on a nonlinear mapping of neural activity measured in multiple timescales, which might incur greater complexity. Therefore, to facilitate the subjects’ learning of the mapping [33,34], we simultaneously adopted auditory and visual feedback in the task.

We found that both subjects exhibited certain improvements in this task based on the trends for the success rate and the trial duration (see “Behavioral Metrics” in Methods for details). The success rate increased over the course of learning (Figure 3A). By further designating the first four sessions as the early phase of learning and the last four as the late phase, we observed marked improvements from the early to the late phase (Figure 3C). In addition, the trial duration decreased (Figure 3B,D). Furthermore, Figure 4 shows that the distribution of NCS was flatter in the task block of the last session (S10) than that of the first session (S1), indicating that the subjects had mastered the modulation of NCS in the late phase of the learning (B11: $var_{S1}$ < $var_{S10}$, *F* = 0.4525, *p* < 0.0001; C05: $var_{S1}$ < $var_{S10}$, *F* = 0.7304, *p* < 0.0001). It is worth noting that the increased success rate and the shortening of the trial duration of successful trials would jointly result in more occurrences of high NCS, which was indeed revealed through the lens of logarithmizing distributions (Figure 4C,D). Collectively, the novel task we designed was learnable, under which the subjects became proficient in modulating the non-linearly mapped NCS.

We next asked whether up-modulating NCS could evoke the precise temporal pattern in this task. To reveal the precise temporal patterns, we applied the cross-correlation histogram (CCH) to the 300 ms window preceding reward delivery (denoted as “Pre-reward”). We first binned the CCH in 15 ms, which was approximately consistent with the time constant of the STDP function. CCH peaked near zero exclusively on the positive side, suggesting a significantly high probability of spike coincidence and a reliable temporal order of the trigger unit leading to the target unit (Figure 5A). In the vicinity of the peak, pronounced troughs were shown, which suggests that temporal precision had been achieved with efficiency rather than via hit-and-miss. Furthermore, since the coincidence score decayed exponentially with increased relative spike timing in the STDP function, the subjects were expected to generate finer temporal patterns (≤5 ms) to improve the chances of obtaining rewards. By narrowing the bin width of the CCHs to 5 ms, we found that the target units indeed fired more compactly with the trigger units than what could be inferred from the coarser CCHs mentioned above (Figure 5B).

On top of the temporal precision, if the elicited patterns between the direct neurons contained fewer spikes, they would be considered efficient temporal patterns; this was expected to be achieved in our BMI paradigm by introducing the normalization to constrain the spike discharging. To test this assumption, we first compared the spike counts from the rewarding neural patterns to those from the baseline block with controlled coincidence scores (see “Comparison of spike counts for precise temporal patterns” in Methods for details). We found that in order to yield neural patterns with identical coincidence scores, the subjects would discharge less while engaging in the energy-constrained BMI task than in the baseline block (Figure 6A). To further support that the decrease in spike counts with the identical coincidence scores was caused by the energy-constrained BMI task, we performed a control BMI experiment (CS-modulation task) where the neural activity was mapped to the mere coincidence score without being normalized by the spike counts. The rewarding neural patterns in this control experiment were used to compare the spike counts. It was also shown that subjects discharged less in the NCS-modulation experiment than in the CS-modulation control experiment to generate neural patterns with identical coincidence scores (Appendix A). Although the control experiment was conducted on one subject (i.e., C05), the gap between the training on the NCS-modulation task and the CS-modulation task was long (∼2 months) and filled with other tasks. With different pairs of direct neurons being modulated, it should be justified as an independent experiment to be compared with the results from the NCS-modulation task on both subjects. Together, the normalization introduced in our novel BMI decoder constrained the discharging rates while preserving the temporal precision to support the generation of efficient patterns.

Although we have demonstrated the generation of efficient temporal patterns via BMI between direct neurons, energy expenditure on a network scale was unclear. Therefore, we investigated the firing rates of indirect neurons that showed significant selectivity in the task or baseline blocks. The modulation index was commonly used to measure relative changes in firing rates under distinct conditions normalized by the average firing level [35,36]. Since the down-modulating of firing rates was more of a concern in our case, significant modulation to reduce discharging events would yield a positive modulation index by our definition. In contrast, increasing firing rates relative to the baseline level would lead to a more negative modulation index (Methods). We found that in the late phase of learning, the overall modulation index of the significantly modulated units was either dominated by a positive modulation index or was counterbalanced to zero (Figure 6B). Put differently, the neurons up-modulating their spiking activity did not dominate the network, thereby not causing a global excitation. Hence, we conclude that energy efficiency was achieved at the population level.

Unlike conventional behavioral paradigms that have encountered difficulty in recognizing neural patterns that are causal to behavioral changes when behaviors are not overtrained, BMI endowed us with the ability to track the evolution of rewarding neural patterns (i.e., the efficient temporal patterns) during learning due to its well-defined contingency between the neural patterns and outcomes. Since the conditioned variable NCS was determined by both the temporal precision and the firing rates of neural activity (estimated by data from the “Pre-reward” period), we depicted the rewarding neural patterns across sessions in a neural state space spanned by the temporal precision and the firing rates of neural activity in order to explore the evolution (Figure 7A). It was revealed that instead of being static in the neural state space, the rewarding neural patterns evolved along a principal axis over the course of learning when using the firing rates of the target unit. However, such an evolution was not observed when examining the firing rates of the trigger unit (Appendix A). Over the course of learning, the change in firing rates of the pair of direct neurons was mainly driven by the target unit (Appendix A, test for the significantly non-zero slope of linear fitting, B11: trigger unit, *p* = 0.354; target unit, *p* = 0.004. C05: trigger unit, *p* = 0.705; target unit, *p* = 4 × 10${}^{-4}$). In addition, we found that the variability of the voyage through the neural state space was correlated with behavioral unsteadiness. After classifying the learning sessions as either unstable or stable according to the variation of the success rate (Methods, marked by red dashed circles in Figure 7A), it was shown that the target patterns from those stable sessions tended to reside in a relatively compact area compared to a vaster area occupied by the unstable sessions in the neural state space. This settled compact area in the neural space was shared by both subjects despite their distinct evolutionary directions (Figure 7B), indicating that a general implementation might exist to support the generation of efficient temporal patterns with stability and reproducibility.

## 4. Discussion

We have mentioned two current blind spots in relation to efficient temporal patterns: the scope of computations that efficient temporal patterns could contribute to and the plausibility of biological brains learning to generate efficient patterns. For the scope of computations, BMI transforms the “stimulus–response” perspectives, allowing diverse and complex computations to be studied. More specifically, this study investigated the capability to generate efficient temporal patterns in order to drive the output. Efficient temporal patterns could be understood beyond the sensory system. Moreover, the user-defined “BMI decoder” allowed us to explicitly impose energy constraints directly onto the spikes (Figure 1C). Previously, interfacing with the biophysical properties of a behavioral paradigm was almost unrealistic. *In silico* simulation has been an alternative tool for studying the efficient codes with energy constraints directly on the spikes (Figure 1B), although this method suffers in terms of its biological plausibility. Therefore, BMI bridges two levels of brain research, providing a framework for studying whether and how biological brains learn to generate efficient patterns.

### 4.1. Comparison to the General Efficient Codes

When the efficient neural code is mentioned, at least two meanings are typically included: maximizing representational capacity [37,38] and minimizing metabolic cost by decreasing the active firing of neurons [8]. Therefore, as long as these two aspects are accounted for, neural codes can generally be seen as efficient regardless of the specific coding schemes (e.g., rate code, temporal code, or phase code). However, in our study, we constrained the reasoning and analysis of efficiency in the dimension of the temporal code, which assumed that the temporal precision of neurons encoded the information. It was beyond the scope of this study to address how the information could be maximally encoded by temporal precision. Instead, we focused on investigating minimizing spikes when yielding temporal precision. This issue has its origin in studies reporting that temporal precision could be coupled with the up-modulation of overall firing rates, mostly in the motor cortex [9,39,40].

### 4.2. Implications to the Information Capacity of Temporal Codes

Our results revealed that the rewarding neural patterns of two subjects traversed in a low-dimensional space (i.e., a line in the 2d neural state space) with different evolutionary directions. However, the trajectories seemed to converge in the neural state space at the final stage of learning, when behaviors were stable (Figure 7). To further confirm that this convergence was not the artifact caused by the gap in the rewarding threshold between the subjects, we corrected the difference in the reward threshold and found that the distribution of overshooting NCS also overlapped in the last session of learning (Appendix A).

This convergence and the inhabited subspace suggest that an optimal neural pattern might exist as an optimal solution to this BMI task. The optimality might be subject to (i) biological constraints on neural structures (some patterns are not biologically feasible to be carried out) and/or (ii) some canonical implementations.

The analysis of precise temporal patterns is mainly based on a framework from digital telecommunications, where binary states from one or more neurons in millisecond resolution comprised the “codewords” that were assumed to represent sensory stimuli [41] or movement commands [42]. On top of this framework, many attempts have been made to estimate the entropy of the spike trains [4,43,44,45]. Typically, *N* binary variables would yield $K=2^{N}$ possible words in a dictionary. This dictionary size grows exponentially with *N*. This scaling effect also applies if consecutive time bins of fewer neurons were to comprise words [45]. According to the optimal neural state space that our results suggested, however, codewords could reside in a low-dimensional space, where the size of codewords C≪K, resulting in much lower empirical information capacity than the theoretical one. However, we believe that a further detailed depiction of this low-dimensional space would help to build empirical priors for a bayesian decoding of spike trains [42,46].

### 4.3. Limitations and Open Questions

Our results showed that in both subjects, target neurons played the leading role in the evolution of neural patterns (Figure 7A). Since we constrained the temporal order of spikes in the rewarded patterns in the definition of the NCS, the identity of the direct neurons might not have been interchangeable, which could have led to different firing profiles. Alternatively, the unbalanced roles might have resulted from a small sample size bias. Therefore, future studies should involve more pairs of direct neurons in more subjects to test these possibilities.

One of the factors that distinguished the BMI decoder used in this study and in other studies was the normalization of the firing rates. Normalization is a general neural computation widely performed in various brain regions and under multiple contexts [47]. However, it was unclear whether the brain had evolved some general mechanisms to implement normalization. We found that the efficient patterns appeared to converge in the final stage of learning (Figure 6B), suggesting that the inhabited area might have delineated the neural patterns that had been generated by the general implementation of normalization. Thus, a more thorough examination of the converged target patterns in other dimensions is needed.

Unlike previous research that characterized how efficient temporal codes correlated with over-trained behaviors or reproducible natural behaviors, BMI paves the way for probing neural dynamics throughout learning since the causal relationship between neurons and outcomes is explicitly defined. Therefore, theoretical moment-by-moment or trial-to-trial errors could be explicitly inferred, which allows for future quantitative tests of spike-based learning models [48] or credit assignment models [49], thus inspiring biologically plausible learning algorithms.

## 5. Conclusions

In this study, we proposed a novel BMI-based task to investigate the implementation of efficient temporal patterns. We demonstrated that subjects could learn the task and volitionally generate efficient temporal patterns in a reproducible manner in M1, a brain area containing rich neural dynamics for directing motor output. In particular, generating these patterns did not invoke global excitation and surplus spikes, which would have increased the metabolic cost. In addition, by constructing neural state space via firing rates and fine-timescale spike temporal profiles, we found that the rewarding neural patterns evolved through learning and displayed an inclination to converge and inhabit in the state space.

As discussed in the last section, in future studies, we need to include more pairs of direct neurons in order to test: (1) whether the target neurons play a leading role in this task, and (2) whether the brain exploits some general mechanisms to implement the normalization.

## Figures and Tables

**Figure 1 brainsci-12-01269-f001:**
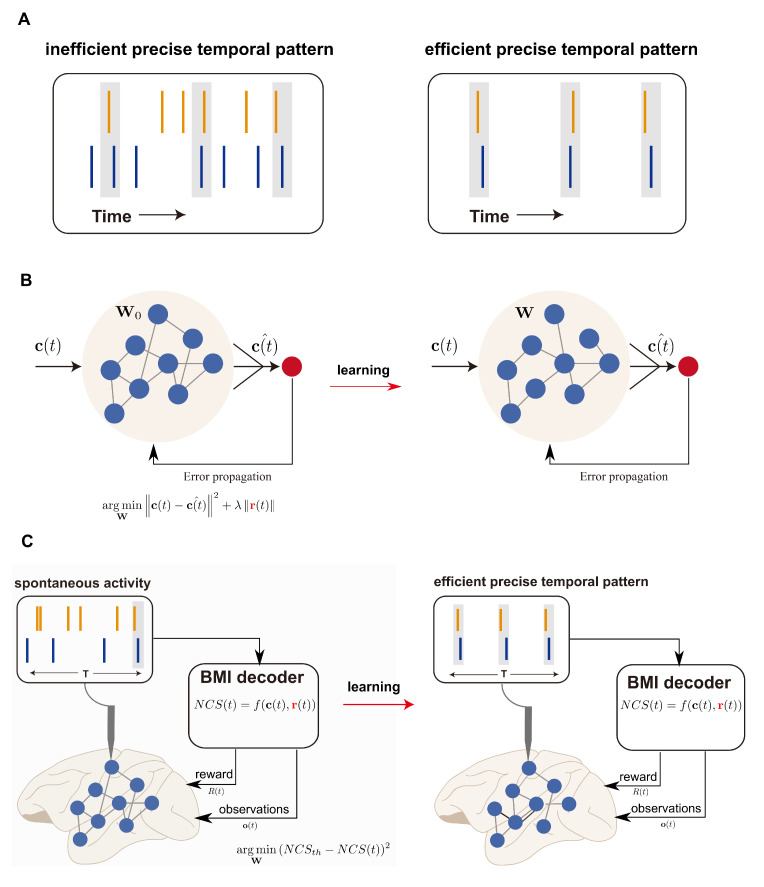
Schematic of precise temporal codes. (**A**) Schematic illustrating two types of precise temporal codes. Shaded rectangles mark the precise synchronization beyond rate fluctuation that is preserved in the spike trains of two neurons. However, the efficient precise temporal pattern depicted on the *right* contains less spikes than that on the *left*, although the number of spiking coincidences under the two cases is identical. (**B**) Schematic of how efficient codes are studied on artificial neural networks by explicitly incorporating the regularization term into the cost function in order to penalize high firing rates r(t). Here, c(t) denotes the target signals to be represented by the network. (**C**) Schematic of the explicit incorporation of information capacity c(t) and metabolic cost (neurons’ firing rates r(t)) in the decoder of the brain–machine interface for the study of the efficient codes in vivo. NCS(t) is the decoded variable defined in this study.

**Figure 2 brainsci-12-01269-f002:**
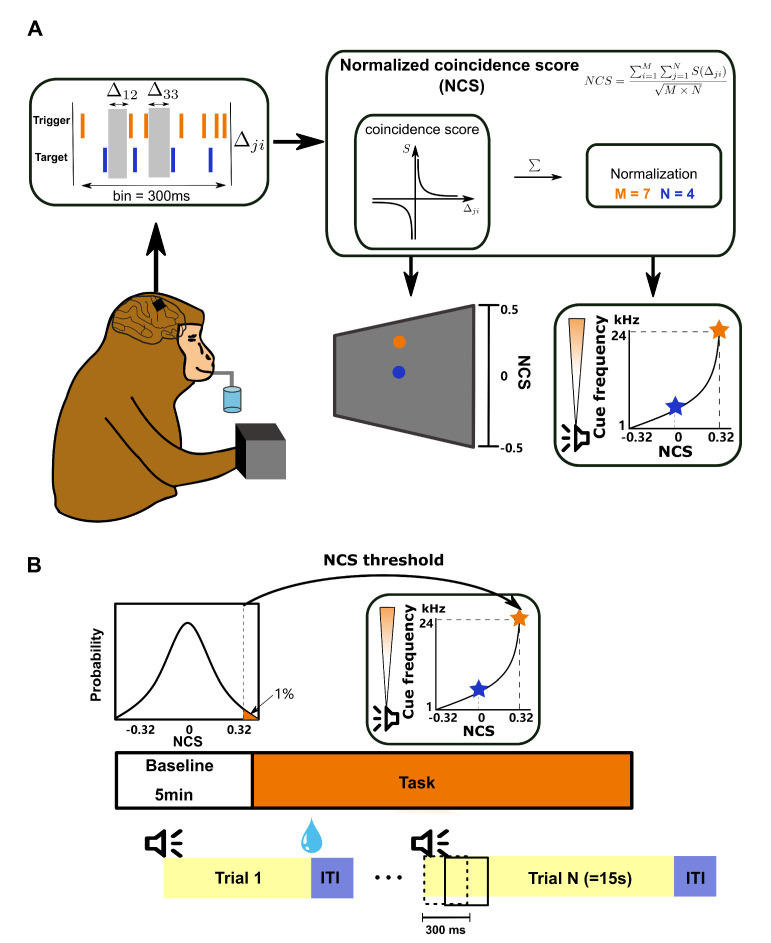
The BMI-based task paradigm. (**A**) The task was run in a closed-loop manner. Neural activity from the subject’s M1 was automatically read and extracted. All leads and lags between spikes from the trigger and target units were used to compute the NCS, which was further fed back by mapping it to the frequency of an audio cursor and the vertical position of a visual cursor. In this exemplified trial, the NCS threshold for reward was 0.32. The threshold for the water reward was set based on the NCS distribution estimated in the first session. (**B**) Task structure of one typical session. The session started with a 5-min baseline block, after which the distribution of NCS could be estimated. One trial could span 15 s at most, followed by a 4 s long inter-trial interval (ITI). The NCS was computed using neural activity in a 300 ms sliding window.

**Figure 3 brainsci-12-01269-f003:**
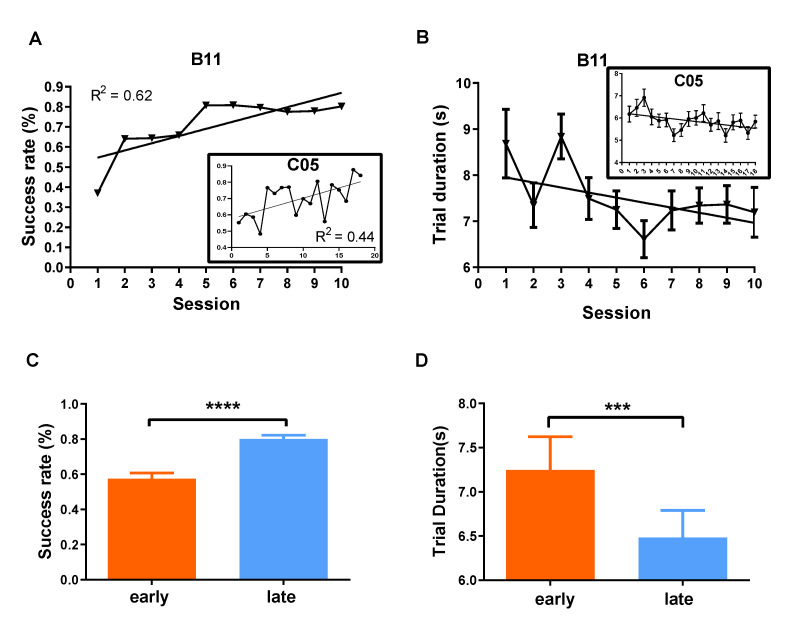
Proficiency increased in performing the BMI-based task. (**A**) The success rate exhibited a growing trend over the course of learning (linear fitting test to determine whether slope was significantly greater than zero: B11: *p* = 0.005; C05: *p* = 0.008). (**B**) The trial duration was reduced over the course of learning (linear fitting to account for sample size N and scatter among replicates and to test whether slope was significantly greater than zero: B11: *p* = 0.048; C05: *p* = 0.010). (**C**) Pronounced increase in the success rate from the early phase to the late phase, pooling over two subjects. One-tailed Mann–Whitney test, early < late, *p* < 0.0001. (**D**) Significant shortening in trial duration from the early phase to the late phase, pooling over two subjects and all trials (Mean ± SEM), F(1,1729) = 12.78, *p* = 0.0004. Interaction between two factors: F(7,1729) = 1.14, *p* = 0.3338. *** *p* < 0.001, **** *p* < 0.0001.

**Figure 4 brainsci-12-01269-f004:**
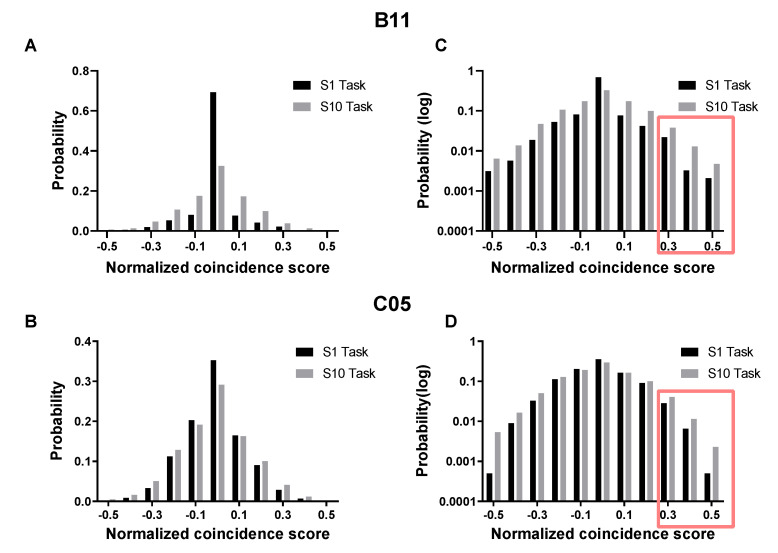
The stronger modulation of NCS in the last session of learning indicates that reproducible neural patterns were generated. (**A**,**B**) The NCS distribution of the task block in the first session (S1) compared to the last session (S10). (**C**,**D**) Distribution on the right side of the panel was shown in the y-axis, logarithmically scaled to better reveal the growing frequency of high NCS, especially the range over the NCS threshold (i.e., 0.3–0.5), as indicated by the red box.

**Figure 5 brainsci-12-01269-f005:**
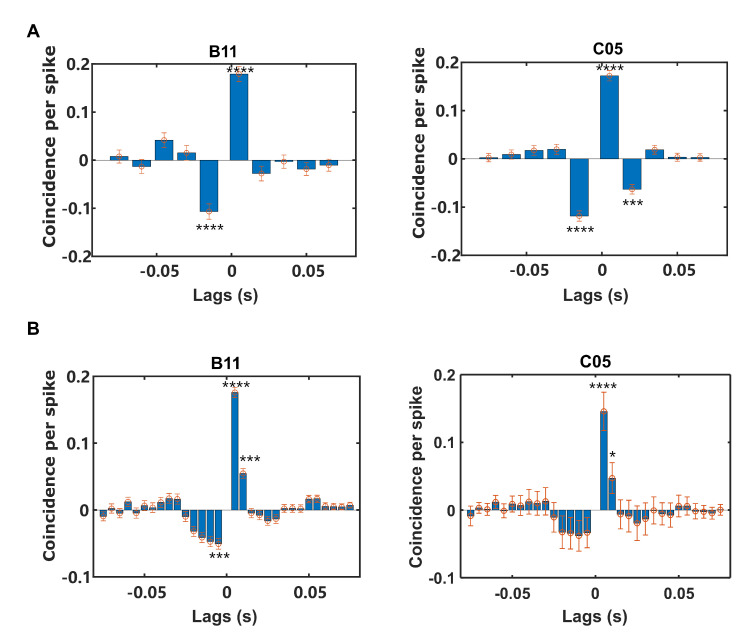
Rewarding neural patterns of BMI-based tasks were temporally precise. (**A**) CCHs of two subjects Attn AE - part of Figure 5 caption in the last session showed pronounced coincidence between two units in a time window spanning from 0 to +15 ms. Error bars denote the SD rendered from the resampling analysis on jittering. The asterisks above the bars indicate a significant coincidence. **** *p* <0.0001. (**B**) The CCHs of two subjects in the last session with higher resolution showed finer temporal granularity. Error bars denote the SD rendered from the resampling analysis on jittering. The asterisks above the bars indicate a significant coincidence. * *p* < 0.05, *** *p* < 0.001, **** *p* < 0.0001.

**Figure 6 brainsci-12-01269-f006:**
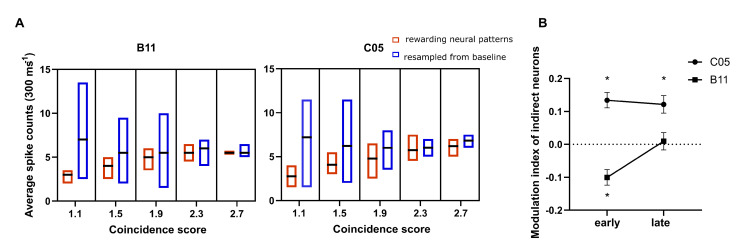
Efficient firing in modulating NCS for the BMI task. (**A**) The spike counts (averaged over trigger unit and target unit) of the rewarding neural patterns from the task block were significantly lower than those of the sampled neural patterns from the baseline block (Mack–Skilling test, B11: *p* < 1 × 10${}^{-4}$; C05: *p* < 1 × 10${}^{-4}$). The spike counts were grouped into five blocks according to the coincidence scores of the corresponding neural patterns. The black line represents the median. (**B**) Average modulation index after pooling over all significantly modulated indirect neurons across sessions (mean ± SEM). The asterisks above the lines indicate a significant positive departure from zero, whereas those below represent values smaller than zero. * *p* < 0.05.

**Figure 7 brainsci-12-01269-f007:**
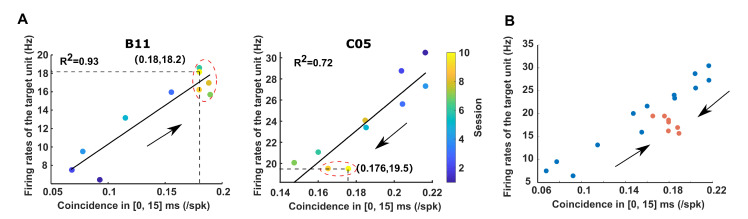
Evolution of rewarded neural patterns throughout learning. (**A**) The CCH coincidence in [0, 15] ms is correlated with the firing rates of the target unit for both subjects (*Left*: B11, *p* = 1.05 × 10${}^{-4}$; *Right*: C05, *p* = 9 × 10${}^{-4}$). However, the evolution showed opposite directions, as indicated by the arrows. The coordinates of the last session are labeled. (**B**) Rewarded neural patterns of both subjects under the same neural space. Each dot represents the averaged neural pattern in one session. Neural patterns from stable sessions are indicated by the *orange* dots.

## Data Availability

Data and codes are available upon reasonable request.

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
