# Peer review of "Volitional Generation of Reproducible, Efficient Temporal Patterns"

_brainsci, 2022, doi:10.3390/brainsci12101269_

Round 1

Reviewer 1 Report

The authors of this work focus on the idea of ​​energy efficiency to encode neuronal activity. To verify this hypothesis, the authors carry out a study where they implement a brain-machine interface where animals receive a reward for increasing the frequency of synchronous activity between two neurons recorded in the primary motor cortex. The main finding reported is related to an increase in the task performance associated with the increase in synchronous patterns without significant changes in the discharge rate of those neurons directly involved in the task.

This work is exciting because it presents a strong hypothesis validated in the results. It is well written, and the results are relevant to both information coding and energy limitations of the brain.

However, before being accepted, it must resolve some minor points.

  1. In English, it is very good, but there are a few confusingly constructed sentences. Ex: Line 6: “In this study, we designed a novel brain-machine interface (BMI) paradigm by learning which two macaques could volitionally generate reproducible energy-efficient temporal patterns in the primary motor cortex (M1). 

  2. The text mentions that the Trigger and Target units were chosen based on their noise signal. The indirect neurons were examined only based on their discharge rate because a correlation analysis was not performed between direct neurons. Did you complete any analysis to determine whether correlation changes also occurred in the indirect ones?

  3. The supplementary figures have legends that are not very informative. For example, figure S1A has several colored curves that indicate what they correspond to. Same as figure s2

Author Response

We appreciate the reviewer for the precious time in reviewing our paper and providing valuable comments. Here is my response to the reviewer's comments.

Point 1: In English, it is very good, but there are a few confusingly constructed sentences. Ex: Line 6: “In this study, we designed a novel brain-machine interface (BMI) paradigm by learning which two macaques could volitionally generate reproducible energy-efficient temporal patterns in the primary motor cortex (M1).

Response 1: We thank the reviewer for pointing out the ambiguous compound expression. We have thus modified it into: “In this study, we designed a novel brain-machine interface (BMI) paradigm. Two macaques could volitionally generate reproducible energy-efficient temporal patterns in the primary motor cortex (M1) by learning the BMI paradigm. ”

Point 2: The text mentions that the Trigger and Target units were chosen based on their noise signal. The indirect neurons were examined only based on their discharge rate because a correlation analysis was not performed between direct neurons. Did you complete any analysis to determine whether correlation changes also occurred in the indirect ones?

Response 2: The reviewer asked “whether correlation changes also occurred in the indirect ones”, which indicated the “correlation analysis” performed on direct neurons but not on indirect neurons. But stating “because a correlation analysis was not performed between direct neurons” at the same time slightly confused us. If we understood it correctly, the reviewer here actually referred to the cross-correlation histogram. In this regard, we agree that further analysis on indirect neurons may help to reveal how network evolved with precise timescales. However, we recorded tens of isolated indirect neurons in each session, which brought about thousands of pairwise CCHs. This is a huge amount to be validly visualized and drawn conclusions from. Often, this type of analysis required more sophisticated algorithms to find reliable correlative groups in fine timescale (e.g., frequent itemset mining introduced in Picado-Muiño et al,.2013 and cell assembly detection by Russo et al., 2017). Therefore, we tend to analyze and discuss this issue in great detail in future studies. In addition, since the BMI paradigm was explicitly designed to induce the precise timing between the direct neurons, we think that the cross-correlation analysis is sufficient to support the hypothesis and validate the design. Therefore, we prefer to treat whether indirect neurons coordinated in fine timescale as another extended topic on top of this article to be discussed in future articles.

Point 3: The supplementary figures have legends that are not very informative. For example, figure S1A has several colored curves that indicate what they correspond to. Same as figure s2

Response 3: The colored curves in figure S1A and figure S2 are the average waveforms of the direct neurons in each session. Given that curves in different colors were greatly overlapped or bundled together, it is justified that direct neurons were stable over the course of learning. We have added the above information into the supplementary figures hoping to clearly illustrate the points. 

Reviewer 2 Report

Undoubtedly, one of the extraordinary characteristics of the biological brain is its low energy expense to implement a variety of biological functions and intelligence compared to the modern artificial intelligence. 

Authors in this paper designed a novel brain-machine interface paradigm, by learning which two macaques could volitionally generate reproducible energy-efficient temporal patterns in the primary motor cortex - M1. 

My comments on the article are as follows:

- I propose to expand the Introduction with more references in the literature on the subject. I think it would be worthwhile to refer to the fact that brain-computer interfaces are also developing dynamically. I suggest referring to the publication: Analysis and Classification of EEG Signals for Brain – Computer Interfaces, Series Title: Studies in Computational Intelligence, Series Volume 852, Springer International Publishing 2020.

- Please explain the use of ANOVA in this article.

- The article should introduce the Conclusions section with a summary and plans for the future regarding the research.

Author Response

We thank the reviewer for carefully reviewing our paper and the pieces of advice. Here's our response to the valuable comments.

Point 1: I propose to expand the Introduction with more references in the literature on the subject. I think it would be worthwhile to refer to the fact that brain-computer interfaces are also developing dynamically. I suggest referring to the publication: Analysis and Classification of EEG Signals for Brain-Computer Interfaces, Series Title: Studies in Computational Intelligence, Series Volume 852, Springer International Publishing 2020.

Response 1: We agreed that it is necessary to emphasize the burgeoning of brain-computer interfaces in the Introduction section, and added the suggested reference (See Line 56).

Point 2:  Please explain the use of ANOVA in this article.

Response 2:  In this article, we compared the trial duration in the early phase and that in the late phase (Figure 3D). However, samples of trial duration from different sessions and subjects were pooled together. These factors might exhibit a certain level of interaction. In order to focus on the main factor (early vs. late) and also assure the interactions between factors won’t be significant, two-way ANOVA was used to process this type of nested structure. We have included the above rationale for using ANOVA in the “Statistical Analysis” subsection of “Methods”. Besides, we also added more detailed statistics to indicate the trivial effect of the interaction between factors in Figure 3D.

Point 3: The article should introduce the Conclusions section with a summary and plans for the future regarding the research.

Response 3: We thank the reviewer for the kind reminder. We have reorganized the Discussion section and introduced the Conclusion section with a summary and future plans.